# Acute Effects of a Speed Training Program on Sprinting Step Kinematics and Performance

**DOI:** 10.3390/ijerph16173138

**Published:** 2019-08-28

**Authors:** Krzysztof Mackala, Marek Fostiak, Brian Schweyen, Tadeusz Osik, Milan Coch

**Affiliations:** 1Department of Track and Field University School of Physical Education in Wroclaw, Poland, Ul. Paderewskiego 35, 51-612 Wrocław, Poland; 2Department of Track and Field, Gdansk University of Physical Education and Sport, ul. Kazimierza Górskiego 1, 80-336 Gdansk, Poland; 3PZLA (Polish Track and Field Association), Mysłowicka 4, 01-612 Warszawa, Poland; 4Athletics Department, University of Montana, Adams Center 32 Campus Drive, Missoula, MT 59812, USA; 5Faculty of Sport, University of Ljubljana, Gortanova ulica 22, 1000 Ljubljana, Slovenia

**Keywords:** sprinting, speed, sprint exercises, step variability, kinematics

## Abstract

The purpose of this study was to examine the effects of speed training on sprint step kinematics and performance in male sprinters. Two groups of seven elite (best 100-m time: 10.37 ± 0.04 s) and seven sub-elite (best 100-m time: 10.71 ± 0.15 s) sprinters were recruited. Sprint performance was assessed in the 20 m (flying start), 40 m (standing start), and 60 m (starting block start). Step kinematics were extracted from the first nine running steps of the 20-m sprint using the Opto-Jump–Microgate system. Explosive power was quantified by performing the CMJ, standing long jump, standing triple jump, and standing five jumps. Significant post-test improvements (*p* < 0.05) were observed in both groups of sprinters. Performance improved by 0.11 s (elite) and 0.06 s (sub-elite) in the 20-m flying start and by 0.06 s (elite) and 0.08 s (sub-elite) in the 60-m start block start. Strong post-test correlations were observed between 60-m block start performance and standing five jumps (SFJ) in the elite group and between 20-m flying start and 40-m standing start performance and standing long jump (SLJ) and standing triple jump (STJ) in the sub-elite group. Speed training (ST) shows potential in the reduction of step variability and as an effective short-term intervention program in the improvement of sprint performance.

## 1. Introduction

Past investigations have indicated a significant commonality between sprinting performance (across distances from 20 to 100 m) and explosive power [1,2,3,4,5]. As the largest inhibitor in the sprinting movement is gravity, sprinters must produce large vertical ground reaction force during step take-off to achieve maximal velocity [6]. This is achieved via application of plyometric training as it uses jump-based movements to train and improve explosive power. However, this training modality does not involve a running component.

A mixture of plyometric characters exercises (skips and bounds) and some speed running drills created an alternative sprint-specific running methodology, which represents a high-intensity speed training (ST). ST is treated as a low-volume training strategy and credited with producing significant gains in maximal running velocity [7]. This form of training is based on short bouts of all-out sprints separated by rest periods from 90 s to 5–6 min to enhance post-exercise recovery and avoid fatigue, including central nervous system fatigue [8,9,10]. Similar in one regard to plyometrics, sprint-specific running also involves a rapid eccentric movement followed by a short amortization phase that is then followed by an explosive concentric movement [6]. This enables the synergistic muscles to engage the myotatic-stretch reflex during the stretch-shortening cycle [11]. Such a training stimulus activates the elastic properties of the muscle fibers and connective tissue, allowing the muscles to store more elastic energy [11,12,13]. This then enhances the release of the accumulated mechanical energy during step take-off, a process which is easily visible during skipping and speed bounding. Hence, improvements in elastic energy storage via ST could be interrelated with improvements in single sprint step performance and, therefore, maximal velocity.

Running velocity is a product of step frequency (SF) and step length (SL) [3,14,15] in which a step defines half a running cycle or the interval between ground contact of one foot to the moment when contact is made by the opposing foot [16]. SF and SL are mutually interdependent in which an increase in SL results in a decrease in SF and vice versa [17,18,19,20,21]. Both variables are determined by anthropometric characteristics, movement regulation processes, motor abilities, and energetic processes [16,22]. Hence, the relationship between SF and SL is unique for each sprinter [19,23]. Previous research investigating the interaction between SL and SF in running has generally been inconclusive. There is much debate as to whether sprinters achieve greater benefits from increased SL or SF. Nonetheless, the strong interdependency between SL and SF and performance at sub-maximal and maximal velocities posits that these are important outcome measures when assessing variability in sprinting technique [3,24]. In addition, little is known on how other kinematic variables such as flight phase time, ground contact time, and step velocity interact and influence maximal running velocity.

The fundamental goal of this study was to investigate the potential effects of acute ST on sprint steps kinematics—how these spatial-temporal characteristics can influence sprint running. A secondary goal was to examine performance including maximal running velocity and power of lower extremity. Sprinters of different competitive levels were recruited to better generalize the results of the study. To mitigate the effects of overtraining, only eight sessions were administered in two 10- and 12-day modules separated with 10 days of recovery. Furthermore, the intervention was administered during the pre-season period to avoid additional confounding variables. A pre-test/post-test design was adopted and included a wide battery of tests and exercises commonly applied in sprint training.

Based on this consideration, it was hypothesized that the information of selecting the relevant biomechanical parameters obtained after high-speed training, may indicate the elite of the sprinters (seniors and 100 m < 10.50 s) will show less variability of the sprinting step in the 20-m sprint from a flying start than the sub-elite sprinters (juniors and 100 m > 10.50 s).

## 2. Materials and Methods

### 2.1. Participants

Seven elite and seven sub-elite healthy male sprinters with a minimum of four to five years of regular sprint training were recruited. Sprinters participating in the experiment were members of the national team. For the purposes of the experiment, they were divided into two groups: Elite and sub-elite. The main division criterion was age (juniors up to 19 years, seniors above) and the personal best times in the 100 m sprint (below 10.50 s—elite). Each participant was medically cleared to participate in sprint training and presented no orthopedic or physiological limitation or injury that could affect sprint performance. All participants had previous experience (minimum six months) in sprint-related strength, speed, and plyometrics training before enrollment. Written consent was obtained after the protocol and procedures were explained in full. Parental consent was also obtained from those individuals under 18 years of age. Sprinters were instructed to maintain their normal intake of food and fluids during the study period. Additionally, they were instructed to avoid any strenuous physical activity 24 h prior to testing as well as refrain from eating 3 h before test commencement. The study design was approved by the Institutional Ethics Committee.

### 2.2. Design

The study was performed over a four-week period during the pre-season (April–May) just prior to the outdoor racing season. All participants had been engaged in a standard sprint training regimen (five to seven days of a weekly microcycle) from October to November until inclusion in the study. This training regimen involved general and specialized fitness exercises designed to enhance sprinting technique, increase maximal velocity, and develop strength, power, and endurance. Additionally, many of the participants competed during the indoor competition season (January–February).

Eight ST sessions were executed in a 10-day and then 12-day module (Table 1). The modules were separated by 10 days of recovery training with minimal load to promote central nervous system regeneration. ST in the first 10-day module was executed every other day and in the 12-day module every third day. The days with training involved the main SST workout and one short supplementary workout (general fitness) with the order reversed in each subsequent training session. Post-workout recovery was also provided and predominately involved massage treatment and low-intensity swimming.

ST focused primarily on developing sprint technique and maximal velocity by various exercises that consisted of various skips, bounds, accelerations, starts, and maximal intensity sprints (Table 2). Training duration was approximately 90 min. This included a 20–30 min warm-up of jogging, stretching, light jumping, skipping, and submaximal accelerations and a 10–15 min cooldown. Training load was progressively amplified by varying task complexity (running distance or number of foot contacts during skip and bound exercises) and by increasing running velocity (from 85–90% to 100%).

### 2.3. Testing Protocol

Maximal running velocity and lower extremity explosive power were measured one day before the intervention and 48 h after the last training session was completed to ascertain acute training effects. Pre- and post-testing was executed at the same time of the day and under identical conditions. No familiarization was provided as all of the testing protocols were well-known to the participants. A warm-up similar to the one used during the training intervention was administered prior to testing (light jogging, stretching, skipping drills, light jumps and bounds, and 30- to 50-m accelerations). The test battery included the countermovement jump (CMJ), standing long jump (SLJ), standing triple jump (STJ), and standing five jump (SFJ) to assess explosive power. After 60 min of rest, the participants performed the 20-m sprint (from the flying start) to evaluate sprint step kinematics and then 40 m (from a standing start) and 60 m (using starting blocks) to determine maximal running velocity.

### 2.4. Assessment of Lower Extremity Explosive Power

The CMJ was used to determine vertical jumping performance. The OptoJump–Microgate optical measurement system (Optojump, Bolzano, Italy) determined contact and flight times with an accuracy of 0.001 s. No restrictions were placed on knee angle during the eccentric phase and the participants were instructed to perform a dynamic double arm swing to attain maximal height. Three trials were separated by 1 min of rest with the highest jump selected for analysis.

Horizontal jump performance was measured in the following order: SLJ, STJ, and SFJ. From an erect position with parallel feet placement, the participant executed the SLJ and was required to land on both feet in the long jump pit without falling backwards. Jumping distance was measured to the nearest 1.0 cm. The starting position for the STJ and SFJ was similar. In these jumps, after one or more arm and leg swings, the participant performed the required number of forward jumps with each step on an alternating leg and was required to land on both feet in the half-squat position on a special jumping mattress. Three trials were executed for each jump and the longest distance to the nearest 1.0 cm was recorded. A 2 min rest was provided between each trial and 5–6 min between each jump modality.

The reliability of the vertical and horizontal jumping tests was measured using intraclass correlation coefficients (ICC). A posteriori analysis obtained correlations of 0.92 for CMJ jump height and 0.93 for SLJ, 0.93 for STJ, and 0.90 for SFJ jump distances. The large coefficients indicate satisfactory test–retest reliability and may be explained by the extensive familiarization of all participants with executing these jumps.

### 2.5. Assessment of Sprint Performance

The 20-m flying start, 40-m standing start, and 60-m block start were performed on an indoor track integrated with the Brower Timing TC-System (Draper, Utah, USA). The photocells were positioned on the track at the start and finish according to the sprint distance [3]. In the 20-m flying start, the sprinter began from a standing start and accelerated as quickly as possible to attain maximal running velocity within a 20 m run-up. Upon reaching the 20 m mark, the sprinter continued to sprint for exactly 20 m at their maximal velocity. This sprint modality had been previously applied in research and is considered sufficient to achieve maximal velocity [3,6,16,25,26]. Two trials were executed and separated by 2 min of rest. The same OptoJump measuring system was used to measure the spatial-temporal characteristics of the first nine running steps at maximal velocity including step length, step frequency, ground contact time, flight time, and step velocity. In a track configuration, the measurement system uses a series of interconnected rods (100 cm x 4 cm x 3 cm) fitted with optical sensors. Each rod (RX bars and TX bars) is fitted with 32 photocells, arranged 4 cm one from another and 0.2 cm above the ground. The rods were distributed along the length and width of the track (20 m x 1.22 m). The device was integrated with a computer for data storage and processing. After completing the 20-m flying start sprint, the participants performed two trials of the 40 m from a standing start and 60 m from a block start. Rest intervals of 4 and 6 min were provided between trials, respectively. The fastest time in each distance was selected for analysis.

### 2.6. Statistical Analysis

Means (*x*) and standard deviations (*SD*) were calculated for all dependent variables. Student’s *t*-test was used to examine pre- and post-test differences in running velocity and jumping performance. Fisher’s least significant difference (LSD) tests were performed post hoc to determine pairwise differences when significant *F* ratios were obtained. Variability in the nine steps was quantified by calculating the *SD* and confidence intervals (95%CI). The associations between the performance variables (sprint times and jump distance/heights) were determined by Pearson product–moment correlations. Additionally, hierarchical cluster analysis using Ward’s method was used to determine the linkage distances among the kinematic characteristics grouped as elementary determinants of sprint velocity. A statistical power of 0.90 was determined satisfactory and an alpha level of 0.05 was accepted as statistically significant (denoted in bold font).

## 3. Results

Table 3 provides the anthropometric and personal bests in the 60 m and 100 m of the elite and sub-elite sprinters.

Height, body mass, and BMI were similar between the groups. The differences between the elite and sub-elite group for age and personal bests were significant (*p <* 0.05).

Table 4 presents the pre- and post-test results in sprint and jump performance. Significant differences were observed in all variables in which jumping distances increased and running times decreased in both the elite and sub-elite sprinters (*p* < 0.05). Sprint step characteristics are presented in Figure 1. In the elite sprinters, contact time (CT) decreased post-test in steps four to seven whereas the lowest value was attained in the ninth step. CT showed a decreasing trend from the first to ninth step. Among the sub-elite sprinters at pre-test, a trend towards increased CT between the first and ninth steps was observed. At post-test, increased CT was observed in steps four to eight. The flight time (FT) in both groups increased post-test from the first to the eighth step. Greater FT was achieved by the sub-elite sprinters at post-test. Step length (SL) in both the elite and sub-elite sprinters showed an upward trend at pre-test, with increased SL from the first to the eighth step, only to become more linear at post-test. Additionally, post-test SL magnitudes increased in both groups. Step frequency (SF) showed an irregular pre-test trend in both the elite and sub-elite sprinters. Following the intervention, SF increased in the elite sprinters, particularly in the last three steps. Changes in step velocity (SV) were more pronounced compared with the other variables in both groups particularly at pre-test (an increase in two consecutive steps followed by a decrease in the next two steps).

Post-test variability in the sprint step characteristics was significant for group and step (*p* < 0.05) (Table 5). The interactions ST × group, SL × group, ST × SL, and ST × SL × group did not show variability except SV and CT (*p* < 0.05). Consequently, the consistent generation of high horizontal velocity in the run-up resulted in greater running velocity with stable SV when sprinting the 20 m distance.

Table 6 presents the pre- and post-test correlation coefficients for the performance measures in the 20-m flying start, 40-m standing start, and 60-m block start sprints and vertical and horizontal jumping tests. At pre-test, the only significant correlation in the sub-elite group was between the 60-m block start and SLJ (*r* = −0.76) and SFJ (*r* = −0.77). In the elite group, a significant relationship was found between the 60-m block start and the 20-m flying start (*r* = 0.81) and 40 m standing start (r = −0.76). No other significant pre-test correlations were found. Post-test analysis revealed correlations between the 20-m flying start and 40-m standing start and SLJ and STJ performance in the sub-elite group. In the elite sprinters, only the correlation between 60-m block start and SFJ performance was significant (*r* = −0.87). Many of the horizontal jump tests (SLJ, STJ, SFJ) were strongly associated with each other, however, no significant pre- and post-test correlations were observed between CMJ with any of the variables in either group.

Hierarchical cluster analysis of the grouped variables is illustrated in Figure 2 (pre-test) and Figure 3 (post-test). Comparison of the two dendrograms did not reveal any differences between the emerging clusters. At both time points, the individual clusters were grouped similarly to form two large characteristic aggregations. This suggests a congruency of the kinematic variables and a relatively loose relationship with no significant effect of one cluster on the other. Additionally, the Euclidean distances of the clusters in the pre-test and post-test clusters were similar.

## 4. Discussion

Significant post-test improvements in 60-m block start times were observed in both the elite and sub-elite sprinters by 1.04% and 1.23%, respectively. Improvements were also noted in 20-m flying start performance by 4.84% and 3.62% in the elite and sub-elite sprinters, respectively. Only marginal improvements (not statistically significant) were found for 40-m standing start performance, in which sprinting time improved by 0.98% in the elite group (Table 5). While indicative that ST has a positive effect on sprint performance, the findings are difficult to interpret due to the lack of similar data in the literature. However, comparisons can be made with studies that examined the effects of plyometrics training on maximal running velocity. In this context, the results of the present study are comparable with those reported by Kreamer et al. [27], Hennessy and Kilty [12], and Mackala and Fostiak [3]. There is strong evidence that plyometrics training enhances the stretch-shortening cycle of muscle to improve elastic energy storage [28] and generate faster and more powerful movements [29]. ST is similar in this regard in that it can likewise activate the elastic properties of muscle fibers and connective tissue to also allow greater elastic energy storage that, after its release, can provide additional impetus during running [6].

An important question in this regard is whether ST is more effective than plyometrics training in enhancing explosive power and maximal running velocity. The data from this study can be compared with previous research that used an identical testing protocol to assess the effects of six sessions of plyometrics training [3]. When compared with this study, ST shows greater improvements in 20- and 60-m flying start sprint times (by 0.5% to 2%, respectively) and horizontal jump performance (STL and STJ). In turn, the plyometrics intervention showed greater increases in standing jump (SJ) and CMJ performance [3]. Some studies reported strong correlations (*r*) from 0.65 to 0.90 between sprinting and drop jump (DJ), SJ, CMJ performance, depending on the sprint distance (20–100 m) and type of jump [12,30,31]. These findings are comparable with the present investigation, in which significant post-test correlations were observed between 60-m block start times and SLJ in the sub-elite (*r* = −0.76) and STJ in the elite (*r* = −0.85) group. Strong correlations were also observed between 20-m flying start times and STJ performance (*r* = −0.85) in the sub-elite group but not in the elite group.

The secondary purpose of the study was to examine the effects of ST on sprint step kinematics (SL, SF, FT, CT, and SV). Analysis of SL in both groups of sprinters was compared with other research where high-performance sprinters were investigated [16,22,32,33,34]. These studies reported that SL increased with running velocity and with sprint distance. In our study, linear increases in SL (between steps two and eight in the elite and steps two and six and also step eight in the sub-elite sprinters) were observed in both groups at both time points. SL increased only in the sub-elite group by approximately 4 cm at post-test with no changes observed in the elite group. SL was maintained at 228 and 230 cm (sub-elite and elite, respectively) in the last three steps (Figure 1). This may be explained by the fact that the flying start involved a 20-m run-up and, therefore, a combined distance of 40 m. Therefore, it is possible that SL was still increasing with each step (build-up phase) that only plateaued at the end of the 20 m sprint distance when maximal velocity was attained. In turn, the changes in SF were less pronounced and remained relatively similar in the sub-elite group but slightly decreased from 4.38 to 4.33 Hz in the elite group. Post-test SF in the elite group was similar between the second and sixth step (difference of only 0.03 Hz), whereas greater variability was observed between the second and sixth step in the sub-elite group at pre- and post-test.

Considering both SF and SL, the elite sprinters presented greater SF and slightly longer SL than the sub-elite sprinters (Figure 1). This result suggests that the improvement in sprint performance via increased running velocity does not demonstrate the classic dependency between SL and SF. This contradicts the study of Bezodis et al. [19], who reported a weak correlation (*r* = –0.192) between SL and running velocity but a strong correlation between SF and running velocity (*r* = 0.886). In turn, Hunter et al. (2004) found a strong correlation between sprint velocity and SL (*r* = 0.73) and only a weak correlation between sprint velocity and SF (*r* = –0.14). While this contradicts the previous finding, it does confirm Delecluse et al. [35] who used regression analysis to find that ca. 85% of variance in running velocity can be explained by variance in SL. Similarly, Mackala [34] examined whether an increase in SF or SL would increase running velocity to find that SL was more strongly associated with running velocity than SF.

Post-test SV increased from 9.10 m/s to 9.26 m/s in the sub-elite and from 9.93 m/s to 9.98 m/s in the elite group. Single sprint step execution in the elite sprinters showed a linear increase across steps one to nine at both pre- and post-test with no changes in SL in step four and seven when compared with earlier steps three and six. In turn, the sub-elite sprinters showed more pronounced variation in SV and SL, which, in turn, may have perturbed the sprinting movement and thus explain the slower running velocity (Figure 1). As SV is a product of reduced CT during the support phase, this may explain the increased running velocity in the elite sprinters. According to Coh et al. [17], and Alcaraz et al. [36] the most important factor in sprint step efficiency is the support phase, especially the ratio between the braking and propulsion phases. Therefore, maximal running velocity can be achieved only if the force impulse is as small as possible during the braking phase and may be possible by positioning the foot of the push-off leg as close as possible to the vertical projection of the body’s center of gravity on the surface. Although this was not measured in the present study, this is the most rational explanation for the increases in running velocity in the 20-m flying start. Hence, the various sprint distances involved in the SST intervention may have increased execution economy during the support phase by positioning the center of gravity closer to the fulcrum upon landing, thereby increasing the velocity of each step.

No significant changes were also observed in CT and FT in either group of sprinters (Figure 1). The difference between CT and FT was ca. 0.04 s and was relatively linear from step two to step eight at both pre- and post-test. More variability was observed in post-test FT in the elite sprinters although the difference between the minimum and maximum values is 0.03 s. CT was reduced in the elite sprinters and was comparable with values reported in other studies during maximal sprinting (90–120 ms) [15,37,38] noted a decreasing trend in CT in the first 10 sprint steps after which CT stabilizes.

To better understand the effects of ST on the spatial and temporal variability of sprinting, hierarchical cluster analysis was applied. Post-test analysis revealed that the variables in the first cluster show greater Euclidean distances between CT and FT (6.5 at pre-test and 7.5 at post-test). Changes were also observed in the grouping order regarding the CT, which are arranged in the order of the executed steps. Post-test changes in the second cluster (SF and SV) revealed closer sub-cluster linkages (2.5 units at pre-test and 2 at post-test). These results suggest that SF and SV show considerable dependency and may be associated with improvements in running velocity. Similar conclusions can be assumed for CT and FT based on the Euclidean distances.

## 5. Conclusions

In summary, this study has shown that the application of eight SST sessions are effective in significantly increasing 60-m block start and 20-m flying start sprint performance. Significant improvements were also observed in lower extremity explosive power as ascertained by vertical and horizontal jump testing. Greater increases were observed in the CMJ (mean 7.95% increase) than in the horizontal jumps (mean 2.5–5.3% increase). Increased 20-m running velocity was associated with increases SV, as CT and FT did not change after SST and only relatively linear increases were observed in SL and SF from step two to step eight.

Additionally, sprint-speed training can be recommended as an effective short-term intervention to improve sprint performance and lower extremity explosive power, particularly when considering the required training volume of eight sessions (across 22 days). Sprint performance gains can also be optimized by decreasing variability in sprint step kinematics during maximal velocity running in both lower and higher performing sprinters.

## Figures and Tables

**Figure 1 ijerph-16-03138-f001:**
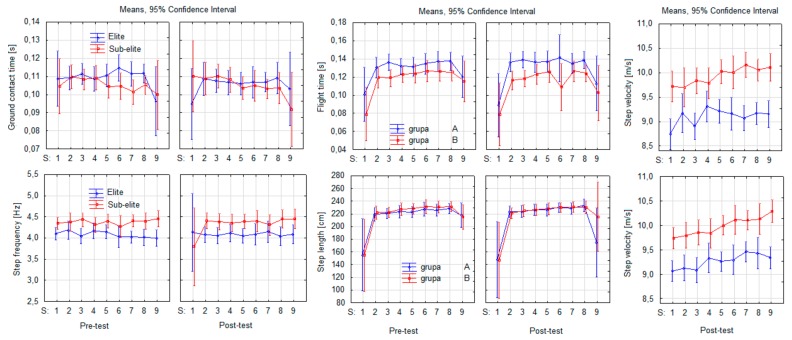
Sprint step kinematics characterizing the first nine steps (S) in the 20-m flying start sprint.

**Figure 2 ijerph-16-03138-f002:**
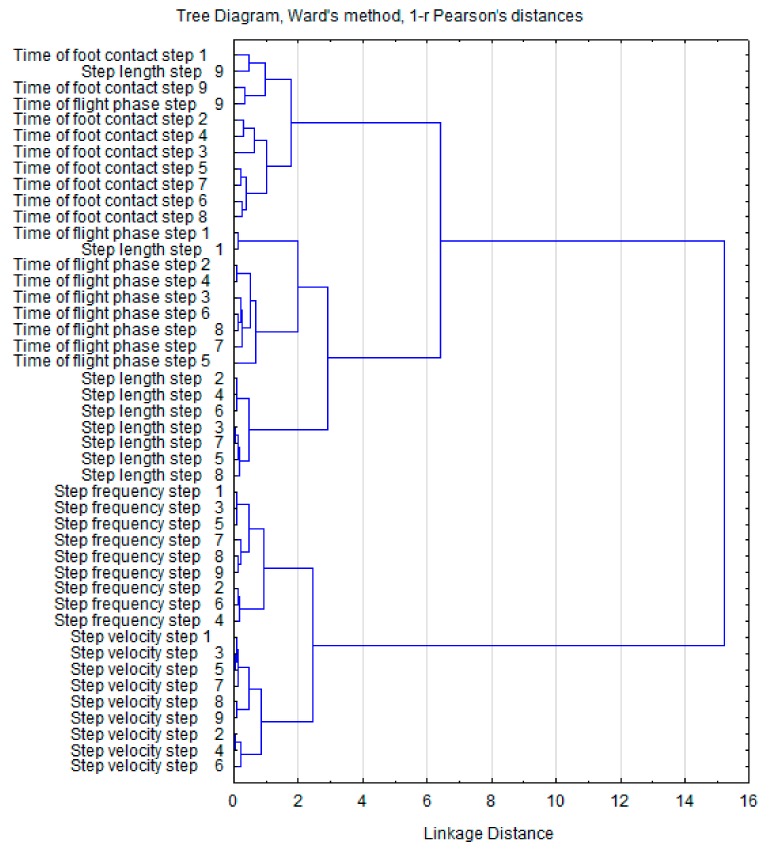
Dendrogram clustering of pre-test sprint step kinematics.

**Figure 3 ijerph-16-03138-f003:**
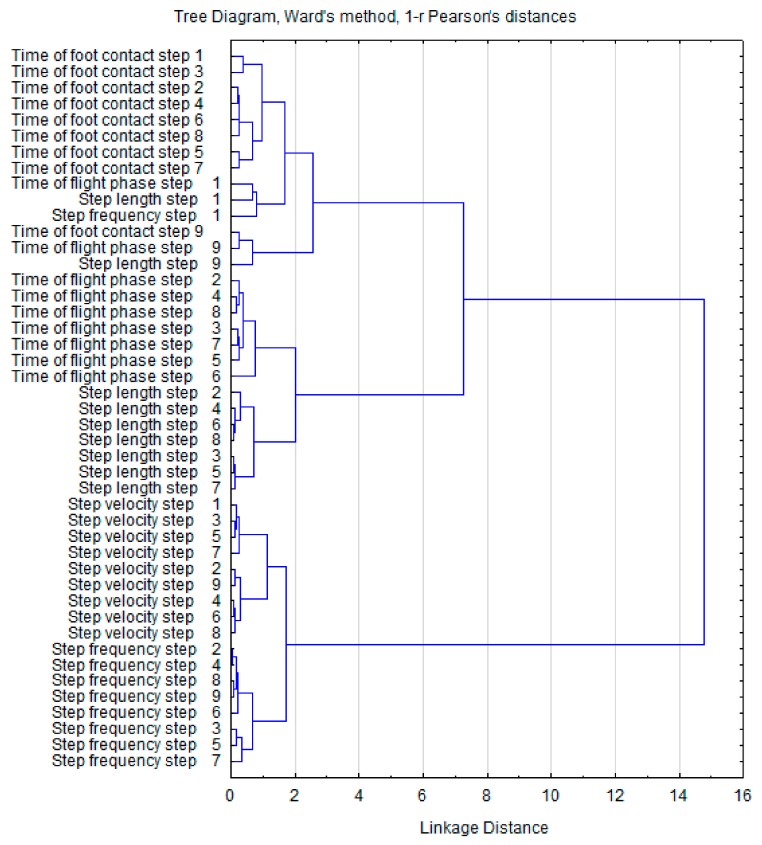
Dendrogram clustering of post-test sprint step kinematics.

**Table 1 ijerph-16-03138-t001:** Training characteristics of the training modules.

Type of Exercise Modality	Training Module	
	10-day Module	12-day Module
	Number of Workouts (*n*)	
Strength (combined with a short plyometrics session)	3	2
Plyometrics	−	1
Speed	4	4
Speed−endurance	1	2
Tempo	2	1
General fitness (supplementary session)	4	4
Recovery (swimming, massage, cryotherapy)	5	5
Day off (rest)	−	1
Testing	−	1
Total training workouts per module	36	

**Table 2 ijerph-16-03138-t002:** Speed training (ST) characteristics.

Exercises	Module 1				Module 2			
	1st session	2nd session	3rd session	4th session	5th session	6th session	7th session	8th session
Sub−maximal speed (85–90%)								
10 m skip A + 20 m acceleration	2 rep.	3 rep.	2 rep.	2 rep.				
10 m skip C + 20 m acceleration	2 rep.	3 rep.	2 rep.	2 rep.				
20 m sprint bounding + 20 m acceleration	3 rep.	3 rep.	3 rep.	4 rep.				
Falling start + 20 m build-up	3 rep.	2 rep.	3 rep.	2 rep.				
Block starts + 20 m build-up	4 rep.	4 rep.	4 rep.	4 rep.				
40 m acceleration	3 rep.	4 rep.	3 rep.	6 rep.				
Total distance [m]	540	600	540	640				
Maximal speed (ca. 100%)								
Falling start + 20 m build up					2 rep.	2 rep.	2 rep.	2 rep.
Block starts + 20 build up					3 rep.	4 rep.	3 rep.	4 rep.
30 m sprint					3 rep.	4 rep.	3 rep.	4 rep.
40 m sprint					3 rep.	4 rep.	3 rep.	4 rep.
50 m sprint					1 rep.	0 rep.	1 rep.	1 rep
Total distance [m]					360	400	360	450

**Table 3 ijerph-16-03138-t003:** Descriptive statistics and Student’s *t*-test results of group age, anthropometric characteristics, and personal best (PB) times.

Variables	Sub-Elite		Elite		*t*	*p*
	*x*	SD	*x*	SD		
Age (years)	18.71	0.75	24.71	2.43	−6.24	**0.000043**
Height (cm)	182.00	5.35	179.42	3.91	0.78	0.449165
Body mass (kg)	73.28	4.49	74.43	8.24	−0.32	0.753007
BMI (kg/m²)	22.17	1.10	22.79	0.74	−1.22	0.244110
60 m PB	6.97	0.08	6.69	0.79	6.52	**0.000028**
100 m PB	10.71	0.15	10.37	0.04	5.71	**0.000097**

Bold format: significant differences.

**Table 4 ijerph-16-03138-t004:** Student’s *t* test results for the dependent variables.

Variable	*x*	*SD*	*x*	*SD*	Δ*x*	Δ*xSD*	*t*	*p*	Confidence −95.00%	Confidence +95.0%
Sub-elite	Pre−test		Post−test							
60m–60m_t2 (s)	7.10	0.09	7.02	0.05	0.08	0.04	4.77	**0.0030**	0.038	0.121
20m flying–20m flying_2t (s)	2.21	0.08	2.13	0.05	0.07	0.05	3.57	**0.0116**	0.024	0.129
40m –40m_2t (s)	4.37	0.04	4.32	0.03	0.06	0.01	7.94	**0.0000**	0.040	0.076
SLJ–SLJ_2t (cm)	2.91	0.06	2.99	0.07	−0.08	0.04	−4.81	**0.0029**	−0.120	−0.039
STJ–STJ_2t (m)	8.56	0.16	8.80	0.18	−0.24	0.07	−8.67	**0.0001**	−.0313	−0.175
SFJ–SFJ_2t (m)	14.90	0.62	15.56	0.53	−0.66	0.13	−13.66	**0.0000**	−0.775	−0.539
CMJ –CMJ_2t (cm)	76.43	4.89	82.71	5.34	−6.29	1.70	−9.76	**0.0000**	−7.862	−4.709
**Elite**	**Pre−test**		**Post−test**							
60m–60m_t2 (s)	6.79	0.08	6.72	0.08	0.06	0.02	7.17	**0.0003**	0.040	0.082
20m flying–20m flying_2t (s)	2.07	0.04	1.97	0.07	0.11	0.07	4.17	**0.0058**	0.047	0.179
40m –40m_2t (s)	4.12	0.02	4.08	0.01	0.04	0.02	4.58	**0.0037**	0.018	0.061
SLJ –SLJ_2t (m)	3.15	0.10	3.23	0.11	−0.07	0.05	−3.92	**0.0078**	−0.118	−0.027
STJ–STJ_2t (m)	9.39	0.52	9.89	0.48	−0.49	0.21	−6.28	**0.0007**	−0.693	−0.304
SFJ–SFJ_2t (m)	15.81	0.44	16.59	0.57	−0.78	0.50	−4.11	**0.0062**	−1.244	−0.316
CMJ –CMJ_2t (cm)	81.57	2.57	87.86	1.07	−6.28	2.06	−8.08	**0.0001**	−8.189	−4.382

_2t—post−test results, bold format—significant differences.

**Table 5 ijerph-16-03138-t005:** ANOVA results of sprint step kinematics.

Feature	Main Effect													
	Group		ST		PT × Group		Step		Step × Group		ST × step		ST × step × group	
	*F*	*p*	*F*	*p*	*F*	*p*	*F*	*p*	*F*	*p*	*F*	*p*	*F*	*p*
CT	0.27	0.6129	4.09	0.0660	2.79	0.1206	2.14	**0.0388**	0.74	0.6540	0.19	0.9918	2.34	**0.0240**
FT	9.05	**0.0109**	0.53	0.4789	0.85	0.3756	10.58	**0.0000**	0.24	0.9812	0.54	0.8220	0.57	0.8016
SF	4.91	**0.0468**	0.33	0.5763	0.73	0.4100	0.90	0.5167	1.68	0.1121	0.89	0.5247	1.13	0.3484
SL	0.61	0.4489	0.32	0.5841	0.08	0.7828	18.23	**0.0000**	0.25	0.9798	0.50	0.8503	0.50	0.8519
SV	56.64	**0.0000**	2.72	0.1249	0.74	0.4080	8.22	**0.0000**	2.06	**0.0475**	0.48	0.8703	0.80	0.6012

SST—sprint-speed training; bold format—significant differences.

**Table 6 ijerph-16-03138-t006:** Spearman rank correlation coefficients between sprint performance and lower extremity explosive power variables.

Sub-Elite							Variable	Elite						
[7]	[6]	[5]	[4]	[3]	[2]	[1]		[1]	[2]	[3]	[4]	[5]	[6]	[7]
-	***−0.77***		***−0.76***	***0.85***	-	-	60 m [1]	-	***0.81***	***−0.76***	-	**−0.87 ***	-	-
-	-		**−0.76 ***	**0.90 ***	-	-		-	-	-	-	-	-	-
-	-	**−0.85 ***	-	-	-	-	20 m flying start [2]	-	-	-	-	-	-	-
-	-	**−0.76 ***	-	-	-	-	40 m [3]	-	-	-	-	-	-	-
***−0.79***	**0.82 ***	**0.79 ***	-	-	-	-	SLJ [4]	-	-	-	-	-	**0.85 ***	-
-	***−0.90***	-	-	-	-	-	STJ [5]	-	-	-	-	-	***0.79***	-
-	**0.79 ***	-	-	-	-	-		-	-	-	-	-	-	-
-	-	-	-	-	-	-	SFJ [6]	-	-	-	-	-	-	-
CMJ [7]														

Italic means pre-test result, bold with ***** means post-test result.

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
