# Peer review of "Acute Effects of a Speed Training Program on Sprinting Step Kinematics and Performance"

_ijerph, 2019, doi:10.3390/ijerph16173138_

Round 1
Reviewer 1 Report
This manuscript sought to compare speed training on sprint step kinetics in elite and subelite sprinters. I think the strength of this manuscript is the methodology, and have a few comments which may assist the authors moving forward.
-No hypothesis is provided.
-Line 51. Why are the the references not in numerical order? See also line 56, 147, 253.
-Does this study have enough power? With only 7 per group I have my concerns. Was an apriori power calculation completed?
-Line 95. What is 'SST', as I have not yet seen that defined.
-What software was used for analysis?
-Table 3. The age is presented with two decimals. Is that an average, or are your methods for quantification of age accurate to the month and day? Also, BMI has no units.
Author Response
Dear reviewer,
Thank you very much for your hard and detailed analysis and evaluation of our manuscript. We followed your comments and suggestions. We have made the necessary corrections. The changes are marked in red. We hope that this has improved the quality of this publication. Once again thank you.
This manuscript sought to compare speed training on sprint step kinetics in elite and su-belite sprinters. I think the strength of this manuscript is the methodology, and have a few comments which may assist the authors moving forward.
-No hypothesis is provided.
We've been thinking hard about the hypothesis. The experiment described in this manuscript is the second part of the overall research problem. In a more recent study (JSCR 2015), we analyzed the impact of plyometric training on improving the power of the lower limbs, maximum speed and, above all, the variability of the kinematic parameters of the sprint step. Publication in JSCR 2015. In this experiment (submitted to IJERPH), of course another group of sprinters, we tried to assess the impact of speed training on improving the power of the lower limbs, maximum speed and, above all, the variability of the kinematic parameters of the sprint step. Therefore, it was difficult to work out the right hypothesis. After analyzing both experiments, the fundamental conclusion is that speed training causes smaller changes in the kinematic variation of the sprinting step rather than plyometric training. This information is prepared in the next manuscript.
In this manuscript, the only logical assumption that can be made is that: It should be assumed that the sub-elite sprinters will show greater variability of the sprinting step than experienced and with high results athletes after applying high-speed training. Therefore we incorporated this hypothesis into text.
-Line 51. Why are the references not in numerical order? See also line 56, 147, 253.
Running velocity is a product of step frequency (SF) and step length (SL) [14,3,15] in which processes [22,16]. Hence, the relationship between SF and SL is unique for each sprinter [19,23]. research and is considered sufficient to achieve maximal velocity [6,16,25,3,26]. were where high performance sprinters were investigated [32,22,16,33,34]. These studies reported that SL
The references are in numerical order. The next in order reference number is marked in green. Numbers in the middle are repeated, previously quoted references.
-Does this study have enough power? With only 7 per group I have my concerns. Was an apriori power calculation completed?
14 sprinters took part in the experiment. This is the national team of sprinters divided into two categories, which differ in age and results. They were juniors as a sub-elite and seniors as an elite. This is a natural division. Both groups of sprinters were at the national training center during the experiment (Spała) and were subjected to similar training - i.e. the impact of high-speed training on kinematic changes of the running step, lower limb power measurements and running speed measurements. The difference between the groups was the volume of training, the greater the volume of seniors. The intensity in speed training should always be at a level close to the maximum. Therefore, the proposed division seems to be a logical and closed division.
-Line 95. What is 'SST', as I have not yet seen that defined.
The days with training involved the main SST workout and one short supplementary - here is a written mistake, too much S, should be ST - speed training
-What software was used for analysis?
We used Statistica software
-Table 3. The age is presented with two decimals. Is that an average, or are your methods for quantification of age accurate to the month and day? Also, BMI has no units.
This is the average age, we added units for BMI (kg/m²)
With regards,
Authors
Reviewer 2 Report
Strengths of paper:
Very well written with very few grammatical errors
Access to high performing / elite athletes
Areas of improvement:
Low sample size
Formatting of Table 4 and Figure 1 (Please refer to specific comments below)
Multiple t-tests utilised (potential for Type 1 error)
Justification of training protocol
Specific comments:
Line 95 - It looks like this is the first occurence of 'SST' - please define
Table 1 - How did you come up with these training modules? Is there a justification for these or a reference from another study?? The types of training modules appear quite random and I'm not clear how they were determined. For example, there was only one plyometric training session in the whole intervention.
You have conducted several t-tests and ANOVA's without any adjustment to the p value. I realise these are on different factors, but one must always be wary of reporting on several tests at the 95% level of confidence without considering the possibility of Type 1 errors.
Table 4 is confusing to me. Has the uploaded document formatted correctly? The headings don't appear to be in the correct place and 'delta mean' appears twice.
Table 4 also seems to present significant findings when means are virtually identical (e.g. please refer to the 3rd row where 4.30 is compared to 4.31 and p is 0.0000).
Figure 1 - Is there are reason why each graph includes Elite and Sub-Elite together rather than Pre-test and Post-test? If you are comparing Pre and Post tests, would it not be more useful compiling that data together on the same graph and Elite / Sub-Elite on different graphs?
Line 233 - Typo --> "There"
Discussion section - Be careful making comments about non-signicant results or results that are statistically significant but have little practical significance.
Author Response
Dear reviewer,
Thank you very much for your hard and detailed analysis and evaluation of our manuscript. We followed your comments and suggestions. We have made the necessary corrections. The changes are marked in red in main text. We hope that this has improved the quality of this publication. Once again thank you.
Strengths of paper:
Very well written with very few grammatical errors
Access to high performing / elite athletes
Areas of improvement:
Low sample size
14 sprinters took part in the experiment. This is the national team of sprinters divided into two categories, which differ in age and results. They were juniors as a sub-elite and seniors as an elite. This is a natural division. Both groups of sprinters were at the national training center during the experiment (Spała) and were subjected to similar training - i.e. the impact of high-speed training on kinematic changes of the running step, lower limb power measurements and running speed measurements. The difference between the groups was the volume of training, the greater the volume of seniors. The intensity in speed training should always be at a level close to the maximum. Therefore, the proposed division seems to be a logical and closed division.
Formatting of Table 4 and Figure 1 (Please refer to specific comments below)
Multiple t-tests utilised (potential for Type 1 error)
We agree, but I wouldn't take it too big a mistake, sometimes more tests are needed, so that you can better understand the data analysis. Some tests better describe existing relationships. Thank you for suggestions and guidance,
Justification of training protocol
We described it below
Specific comments:
Line 95 - It looks like this is the first occurence of 'SST' - please define
The days with training involved the main SST workout and one short supplementary - here is a written mistake, too much S, should be ST - speed trainingTable 1 - How did you come up with these training modules? Is there a justification for these or a reference from another study?? The types of training modules appear quite random and I'm not clear how they were determined. For example, there was only one plyometric training session in the whole intervention.
The experiment described in this manuscript is the second part of the overall research problem. In a more recent study (JSCR 2015), we analyzed the impact of plyometric training on improving the power of the lower limbs, maximum speed and, above all, the variability of the kinematic parameters of the sprint step. Publication in JSCR 2015. In this experiment (submitted to IJERPH), of course another group of sprinters, we tried to assess the impact of speed training on improving the power of the lower limbs, maximum speed and, above all, the variability of the kinematic parameters of the sprint step. As I mentioned early the sprinters were the national team divided into two categories: juniors as a sub-elite and seniors as an elite. Training modules are based on training that was carried out on two sports training camps, carried out one after the other with a 10-day break. This is how Polish sprinter training works. Because I am also sprint coach with international experience (Poland , Spain, Canada, USA), for experiments, the trainers of both groups agreed to modified their trainings so that both groups had similar training. As you can see one module applies to technical speed training (intensity up to 88%) the other module is the maximum speed (maximum intensity). Because the main scope of the module was application of speed training only one plyometric session was applied. I hope this is cleared for you.
You have conducted several t-tests and ANOVA's without any adjustment to the p value. I realise these are on different factors, but one must always be wary of reporting on several tests at the 95% level of confidence without considering the possibility of Type 1 errors.
Table 4 is confusing to me. Has the uploaded document formatted correctly? The headings don't appear to be in the correct place and 'delta mean' appears twice.
You are right. We have corrected the table. Above the measurements we placed a group inscription and the pretest and post-test was placed in the right position middle. Indeed, Delta was placed twice, the second Delta regards SD. We corrected it. Please see the table,
Table 4 also seems to present significant findings when means are virtually identical (e.g. please refer to the 3rd row where 4.30 is compared to 4.31 and p is 0.0000).
We agree, We checked the table and found an error you marked. We rewrote the columns wrong. We corrected the inaccuracies.
Figure 1 - Is there are reason why each graph includes Elite and Sub-Elite together rather than Pre-test and Post-test? If you are comparing Pre and Post tests, would it not be more useful compiling that data together on the same graph and Elite / Sub-Elite on different graphs?
We thought about it, but there would be more tables to publish. A large number of scientific journals limit the number of tables and charts. Showing everything in one graph, but regarding only one factor seems logical and does not disturb the analysis. We have a comparison of both groups and a comparison of how the values of the particular factor changed in both groups before and after.
Line 233 - Typo --> "There"
We corrected it
Discussion section - Be careful making comments about non-signicant results or results that are statistically significant but have little practical significance.
Thank you very much for this observation and suggestion. I think that this is not only our fault, but most authors who wanted the results to be clearer, under which lies better readability for the reader.
With regards,
Authors